# Estimation and Forecasting of Rice Yield Using Phenology-Based Algorithm and Linear Regression Model on Sentinel-II Satellite Data

**Abid Nazir [1,*]**, **Saleem Ullah [1]**, **Zulfiqar Ahmad Saqib [2,3]**, **Azhar Abbas [4,*]**, **Asad Ali [5]**, **Muhammad Shahid Iqbal [1]**, **Khalid Hussain [6]**, **Muhammad Shakir [1]**, **Munawar Shah [1]** and **Muhammad Usman Butt [7]**

1  Department of Space Science, Institute of Space Technology, P.O. Box 2750, Islamabad 44000, Pakistan; saleem.ullah@mail.ist.edu.pk (S.U.); shahid.iqbal@mail.ist.edu.pk (M.S.I.); m.shakir@mail.ist.edu.pk (M.S.); munawar.shah@mail.ist.edu.pk (M.S.)
2  Institute of Soil and Environmental Sciences, University of Agriculture, Faisalabad 38040, Pakistan; zulfiqar.dasti@uaf.edu.pk
3  Agricultural Remote Sensing Laboratory (ARSL), National Centre of GIS and Space Application (NCGSA), University of Agriculture, Faisalabad 38040, Pakistan
4  Institute of Agriculture and Resource Economics, University of Agriculture, Faisalabad 38040, Pakistan
5  Department of Applied Mathematics and Statistics, Institute of Space Technology, P.O. Box 2750, Islamabad 44000, Pakistan; asad.ali@mail.ist.edu.pk
6  Department of Agronomy, Faculty of Agriculture, University of Agriculture, Faisalabad 38040, Pakistan; khalidkhanuaf@gmail.com
7  Sustainable Rice Production, Galaxy Rice Mills Pvt Ltd., Gujranwala 52230, Pakistan; usmanbutt9200@gmail.com
*  Correspondence: gee2018abid@gmail.com (A.N.); azhar.abbas@uaf.edu.pk (A.A.)

**Abstract:** Rice is a primary food for more than three billion people worldwide and cultivated on about 12% of the world's arable land. However, more than 88% production is observed in Asian countries, including Pakistan. Due to higher population growth and recent climate change scenarios, it is crucial to get timely and accurate rice yield estimates and production forecast of the growing season for governments, planners, and decision makers in formulating policies regarding import/export in the event of shortfall and/or surplus. This study aims to quantify the rice yield at various phenological stages from hyper-temporal satellite-derived-vegetation indices computed from time series Sentinel-II images. Different vegetation indices (viz. NDVI, EVI, SAVI, and REP) were used to predict paddy yield. The predicted yield was validated through RMSE and ME statistical techniques. The integration of PLSR and sequential time-stamped vegetation indices accurately predicted rice yield (i.e., maximum $R^2$ = 0.84 and minimum RMSE = 0.12 ton ha$^{-1}$ equal to 3% of the mean rice yield). Moreover, our results also established that optimal time spans for predicting rice yield are late vegetative and reproductive (flowering) stages. The output would be useful for the farmer and decision makers in addressing food security.

**Keywords:** rice yield; vegetation indices; hyper-temporal data; PLSR

## 1. Introduction

The rapid increase in the world population exerts pressure on the agriculture sector and threatening the food security of the world [1]. Among cereals, rice is one of the prime sources of food with high nutritive value (i.e., containing carbohydrate, vitamins (B, E, thiamine), and minerals (Ca, Mg, Fe). Rice is widely grown, consumed globally (i.e., daily food of 3.5 billion people worldwide), and accounts for 19% of the dietary energy [2]. Globally, 90% of the rice comes from Asia, which is approximately 640 million tons per annum [3,4]. Pakistan ranks 11th at the global rice production list and contributes 8% to the world's total rice trade [3]. Pakistan produced seven (7) million tons of rice in the year

2017–2018 and earned a foreign exchange of two-billion dollars ($USD) from rice export [5]. The high-quality nutritious rice (e.g., basmati) produced in the country is available at affordable price in the international market and thus contributing in food security for the increasing global population [6].

Timely and accurate predictions of crop yield before harvest at a large scale is critical for food security and administrative planning, especially in the current continually changing global environment and international situation [7,8]. Different approaches have been adopted for precise yield estimation and each method has its own strengths and limitations. For instance, the traditional field surveys and crop statistics are useful for precisely estimating crop yield; however, when crop yield prediction of the large region is desired, the surveys prove inadequate due to budget, time, and large skilled manpower constraints [9]. The use of Earth observation data (remote sensing) offers an effective system for monitoring agriculture and quantifying crop yield at large spatial extent. The remotely sensed solution is fast, cost-efficient, and non-destructive [10,11]. In addition, the repetitive data acquisition capability of remote sensing sensors makes them an ideal choice for retrieving temporal information of crop phenology, plants health (stress), response to weather and soil nutrients (i.e., manure and fertilizer), variation in plant biomass, and ultimately its effect on yield production [12,13].

Satellite remote sensing also enables crop yield estimation at field, landscape, and regional scales for making policies and ensuring food security [14,15]). Yield estimation of various crops, such as wheat [11], corn [16], and sugar beet [17] is done successfully using assimilation algorithms on RS data. In recent research, two approaches are commonly used for this purpose: one is canopy reflectance data, and the other is based on the spectral indices. The free availability of optical remote sensing data of Sentinel-2 satellite with multiple spectral bands in the red, red edge, and near infrared (NIR) is making RS an ideal choice for monitoring agricultural crops, vegetation phenology [18] (Caballero et al., 2020), temporal variability in cropping [19], as well as environmental monitoring and land cover mapping [20].

Different vegetation indices (VIs) derived from satellite images are effective indicators of vegetation status and are positively correlated with crop yield. The Normalized Difference Vegetation Index (NDVI) has been widely used for predicting crop yield and identifying growth stages [20–22]. Similarly, other variants of NDVI such as Soil Adjusted Vegetation Index (SAVI) and Enhanced Vegetation Index (EVI) have been found effective for crop growth monitoring in the initial filling stages of the crop [21,22]. While using canopy reflectance data to directly estimate crop yield, multivariable analysis methods are commonly introduced to support the dataset analysis. Partial least squares regression (PLSR), stepwise multiple linear regression (SMLR), artificial neural network (ANN), etc. are helpful to construct and validate the multivariate remote sensing models of estimating the yield and improve the accuracy of crop yield estimation through satellite remote sensing, specifically when analyzing the quantitative relationship between RS variables obtained from satellite images and crop yield [11].

The impacts of different phenological and growth stages (i.e., vegetative, reproductive, and ripening) on yield production are rarely explored. Very few researchers quantified the impact of growth stages on crop yields and assigned different weightage to different growth stages [21,23]. Some studies conclude that physiological status (e.g., crop growth) and biochemical contents (e.g., nitrogen) of pre-heading stage is more crucial [24,25], while other found that high rice biomass at post-heading stages is essential for optimum production [23]. Similarly, few studies highlight the relationship between late reproductive growth period and rice yield [26,27]. The overall objective of this study is to forecast rice yield and investigate the relationship between remote sensing derived VIs at different phenological stages of rice crop and its yield. The study also aims to identify the most critical growth period (phenological stage) for quantification of rice yield with hyper-temporal sentinel-II and derived-indices using Partial Least Square Regression (PLSR) model to improve the estimation accuracy of rice yield by remote sensing.

## 2. Materials and Methods

### 2.1. Study Area

This study was carried out at Sheikhupora district (Latitude $31°30'00''$ to $31°65'30''$ N and Longitude $73°40'00''$ to $74°23'00''$ E) of Punjab, Pakistan (Figure 1). Sheikhupora district is located between Ravi and Chenab rivers and irrigated by river water from two canals (Upper Chenab and Khanpur Canals). Climatically the region is dominated by the wet monsoon, thus making it favorable for rice crop. The annual precipitation ranges from 120 to 720 mm, which mainly occurs between July and August [28]. The study area is dominated by alluvial clay and loamy soil rich in humus and mineral composition. The mineralogical, chemical and geotechnical compositions of soil (pH = 8, EC = $1.1-4.5$ dS m$^{-1}$ with soil field capacity from 45–71%) make the region ideal for rice cultivation [29]. Due to the favorable conditions, Sheikhupora district is the second largest rice-producing district in Pakistan with an average production of 2–2.5 million tons annually [5,30].

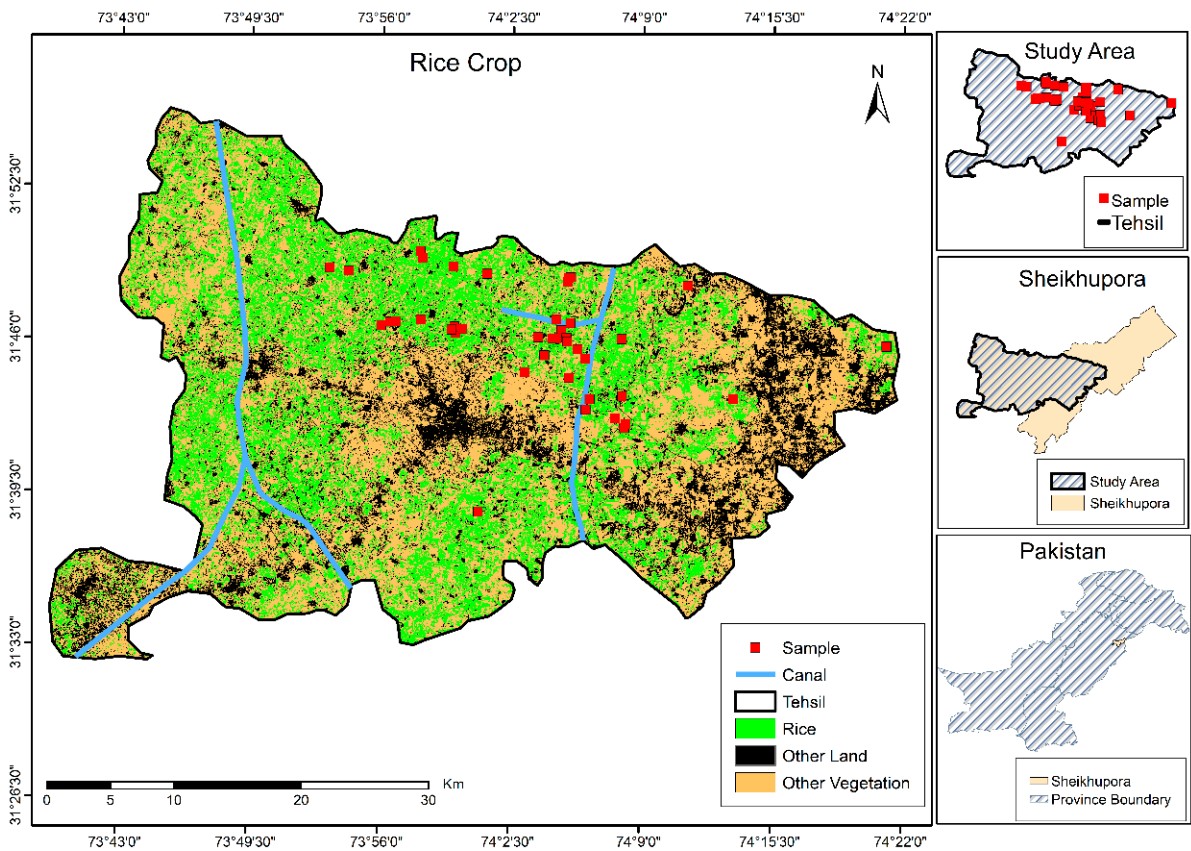

**Figure 1.** Map of the study area (sampling sites are marked as red square dots).

### 2.2. Data Collection

2.2.1. Field Data

Stratified-random-sampling procedure was used to collect data from 137 plots well distributed in the study area. The rice fields with minimum size of $60 \times 60$ m (i.e., corresponding to the coarse pixel size of satellite images used in this study) were considered for the purpose of sampling, monitoring, and analysis. In most of the paddy fields, the rice crop was transplanted in start of July (after seed sown in the nursery at the start of June). These sampled plots were carefully monitored from transplanting till harvesting period. The rice produced in each plot (Figure 2) was carefully measured and recorded for further analysis.

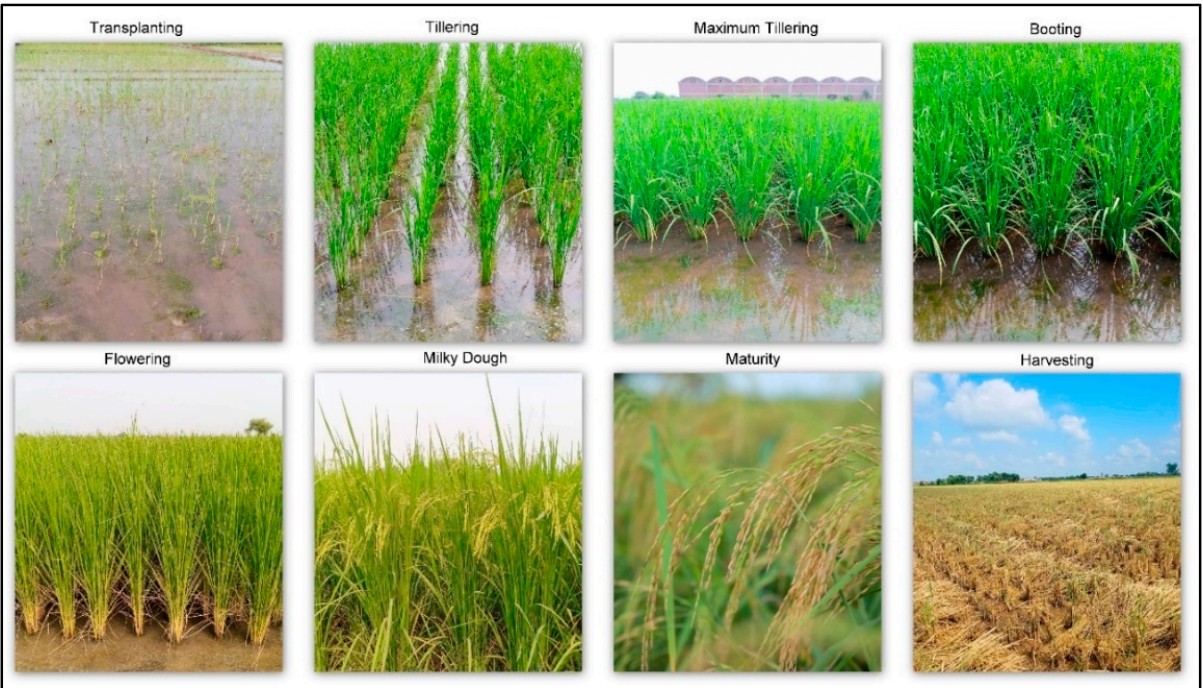

**Figure 2.** The pictorial view of different phenological or growth stages of rice in field conditions.

### 2.2.2. Satellite Data Preprocessing and Vegetation Index

The optical remote sensing data of Sentinel-II satellite for year 2016 was used in this study. The availability of multiple spectral bands in the red, red edge, and near infrared (NIR) part of the EM spectrum with 10–20 m resolution makes Sentinel-II an ideal choice for studying vegetation phenology and monitoring agricultural crops for stress level, nutrient contents, pest attack, and yield estimation [31–33]. The cloud free Sentinel-2 time-series images (i.e., spanning from growing to harvesting phase) were pre-processed for atmospheric correction using Sen2Cor processing. The atmospherically corrected images were then used for computing vegetation indices (Table 1) from times series satellite images spanning from sowing till harvesting period of the rice crop. Vegetation indices are mathematical transformations using two or more spectral bands devised to enhance certain characteristics of vegetation [34]. Several images to cover the entire growth period of rice crop were used and vegetation indices were computed using Google Earth Engine (GEE).

**Table 1.** Mathematical formulas of vegetation indices (NDVI, SAVI, EVI, and Red Edge Position).

| Vegetation Indices (VIs) | References |
|---|---|
| $NDVI = \frac{\rho(NIR) - \rho(Red)}{\rho(NIR) + \rho(Red)}$ | Rouse et al. (1974) [35] |
| $SAVI = 1 + L\frac{\rho(NIR) - \rho(Red)}{\rho(NIR) + \rho(Red) + L}$ <br> where $L = 0.5$, to minimize the brightness effect of soil | Huete (1988) [36] |
| $EVI = G\frac{\rho(NIR) - \rho(Red)}{\rho(NIR) + C1 \times \rho(Red) - C2 \times \rho(Blue) + L}$ <br> where $G = 2.5$; $L = 0.5$ (Soil adjusted factor); $C1$ and $C2$ are constants to reduce aerosols effects. | Liu and Huete (1995) [37] |
| $Red\ Edge = \frac{\rho(Red) + \rho(Red\ Edge3)}{2}$ <br> $REP = 704 + 35\left[\frac{Red\ Edge - \rho(Red\ Edge1)}{\rho(Red\ Edge2) - \rho(Red\ Edge1)}\right]$ <br> where 704 and 35 represent interpolation constants that can be adjusted according to available band's wavelength | Filella and Penuelas (1994) [38] |

### 2.3. Geo-Statistical Analysis

The PLSR is an established multivariate analysis technique commonly used in chemomatric and hyperspectral data analysis [39,40]. The Partial Least Square Regression (PLSR) analyses were performed to all time-series vegetation indices (explanatory variables) with rice yields (response variable). In PLSR model development, the selection of optimum number of latent variables (LVs) is more critical, as increase of the number of LVs would improve the accuracy of the model, while selection of too many variables can lead to the over fitting and the error would increase [41]. To minimize this over fitting problem, the optimal number of LVs was selected based on achieving a combination of a high $R^2$ and a low root mean squared error of the prediction (RMSEP) (Figure 3). The PLSR model was evaluated by plotting the 1:1 relationship graph between the predicted and measured values of the yield (Figure 3). The evaluation indices were the $R^2$ and the RMSE. A larger $R^2$ shows that the model is better, while smaller RMSE values indicate the stronger estimation ability of the model. To evaluate the performances of the prediction models, leave-one-out cross-validation [42] was used, in which the model was iteratively trained on multiple time series data and then used to predict yield. PLSR can be mathematically expressed as:

$$Y = a + b_1X_1 + b_2X_2 + \cdots + b_nX_n \tag{1}$$

where Y is response variable (rice yield), $X_1$, $X_2$–$X_n$ are the selected latent variables (LVs), which are the time series images, in this case a is the intercept and $b_1$–$b_n$ represent the regression coefficients (also known as β-coefficients) for different predictors.

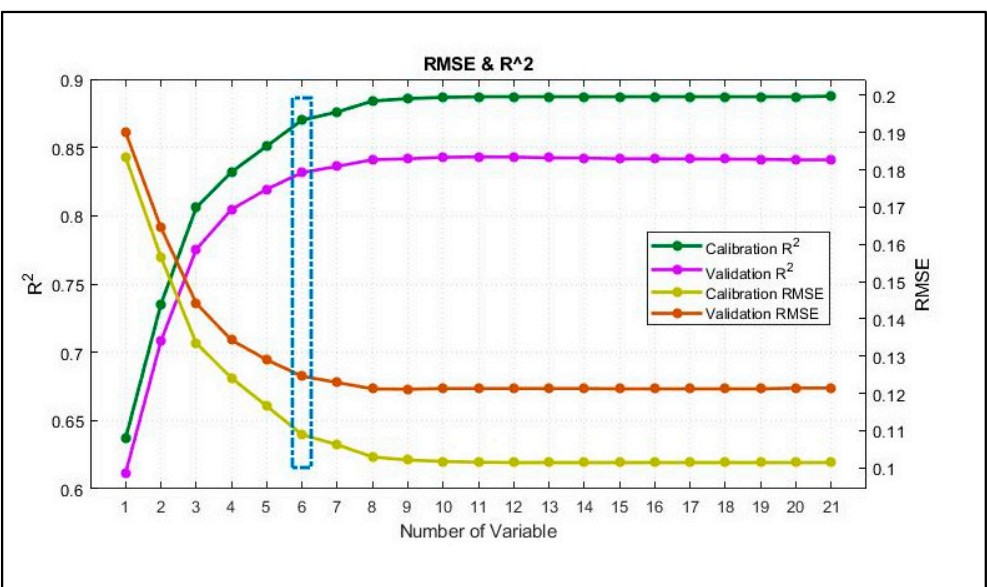

**Figure 3.** The RMSE and $R^2$ plot. The RMSE is decreasing as the number of latent variable increases. After a certain number of latent variables, the decrease in RMSE was negligible and that was taken as the optimum number of variables for PLSR model development.

### 2.4. Spatial Distribution and Mapping of Rice Yield

To model the spatial distribution of rice yields, a two-step procedure was adopted. The thematic map of rice crop was developed using phenological based mapping algorithms (Figure 4). In this routine, the phenology profiles (or signatures) serves as numerical key for discerning different crop types grown in the region of interest [43,44]. The time series vegetation indices (e.g., NDVI profiles computed from optical data for the entire growth span of rice crop: 130 days) were used to demarcate the rice crop and to compute the rice cultivated area. The phenological-mapping-routine takes into account the entire range (minimum–maximum) of vegetation index values (e.g., NDVI

in this case) at each time stamp (e.g., from transplanting till harvesting) and can be mathematically expressed as:

$$R = (NDVI^1min \text{ and } NDVI^1max) \text{ and } (NDVI^2min \text{ and } NDVI^2max) \; NDVI^nmin \text{ and } NDVI^nmax) \qquad (2)$$

where R represents response variable (rice), and $NDVI^1$, $NDVI^2$, and $NDVI^n$ represent ND-VIs derived from RS data at different crop phenological stages starting from transplanting till harvesting.

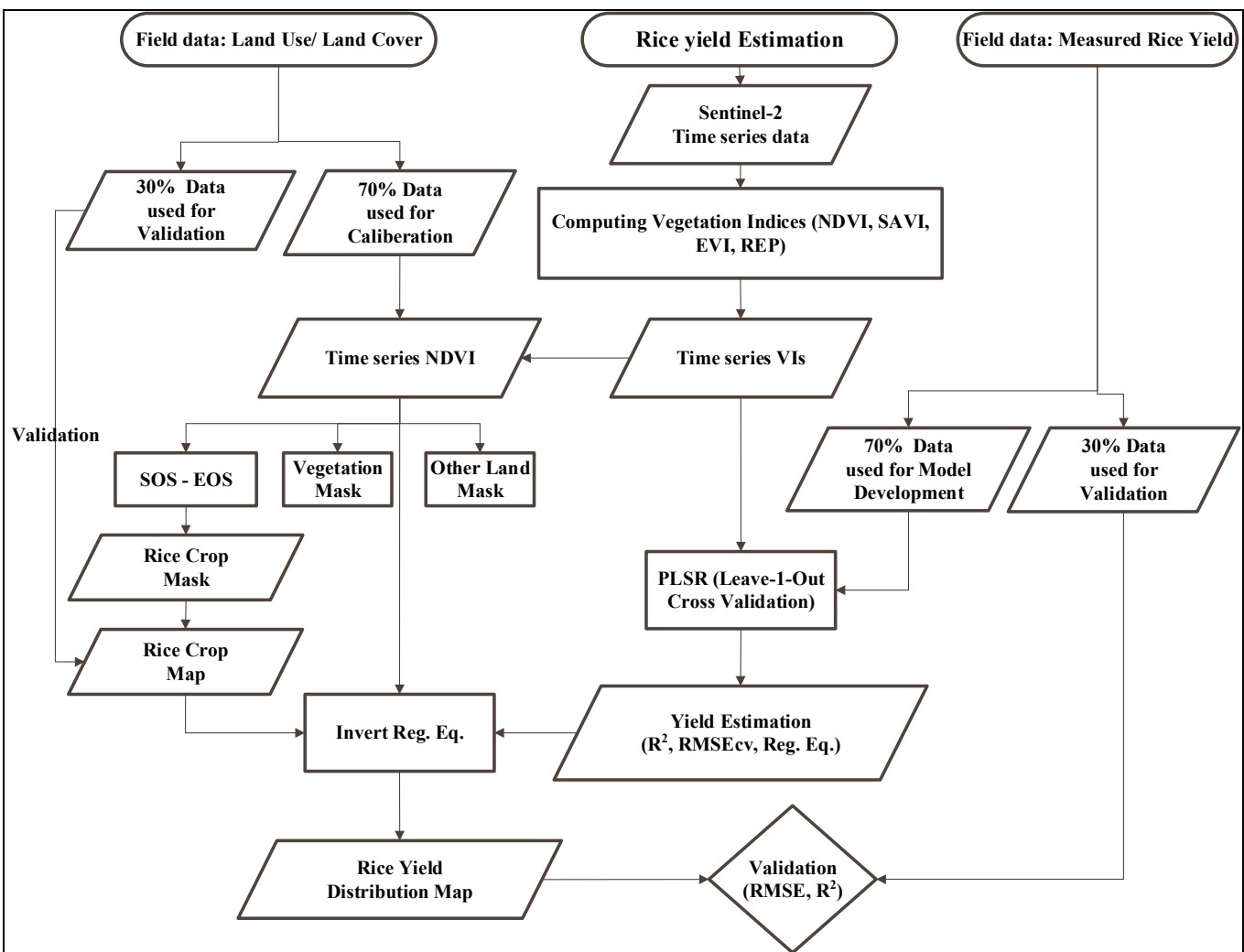

**Figure 4.** The framework of methodology followed for phenology-based rice mapping and yield estimation.

To develop the spatial distribution maps of rice yield, the models of statistical analysis (the regression equations of PLSR analysis) were inverted to the time series vegetation indices of rice masked areas (developed from phenological based mapping) and were validated with an independent dataset.

### 3. Results

#### 3.1. Rice Yield Estimation (Field Level Data)

The statistical descriptions of the measured rice yield are summarized in Table 2. In calibration datasets (with sample size $n$ = 96), the rice yield varies between 3.06 ton/ha and 4.15 ton/ha with mean equal to 3.70 ton/ha. The standard deviation (SD) was ±0.31 ton/ha and coefficient of covariance (CV) equal to 0.083 ton/ha. The graphical display reflects that the calibration dataset is near normally distributed.

**Table 2.** Statistical description of the field measured rice yield with graphical displays of calibration and validation datasets.

| Dataset Type | Sample Size ($n$) | Minimum (ton/ha) | Maximum (ton/ha) | Mean (ton/ha) | SD (ton/ha) | CV (ton/ha) | Graphical Distribution |
|---|---|---|---|---|---|---|---|
| Calibration | 96 | 3.06 | 4.15 | 3.70 | 0.31 | 0.083 |  |
| Validation | 41 | 3.16 | 4.15 | 3.71 | 0.29 | 0.078 |  |

The summary statistics of validation sets ($n$ = 41) shows that rice yield ranges between 3.16 ton/ha and 4.15 ton/ha with average equal to 3.71 ton/ha. The standard deviation (SD) was ±0.29 and coefficient of covariance (CV) equal to 0.078 ton/ha. The validation data are also normally distributed (Table 2: see the graphical display at last column).

#### 3.2. Variation in Temporal Profiles of Vegetation Indices with Rice Phenology

The temporal profiles of vegetation index spanning across the full growth period of rice crop are shown in Figure 5. The vegetation index (e.g., NDVI) values were least (minimum) at transplanting phase and showed gradual increase with increase in vegetative parts (Figure 5). The vegetation indices reached at peak in the late vegetative phase and continually maintained high values (e.g., spectral plateau) till flowering phase. At post flowering phase (e.g., ripening phase), the vegetation index values started declining and reached its minimum at fully ripened harvesting phase (see Figure 5).

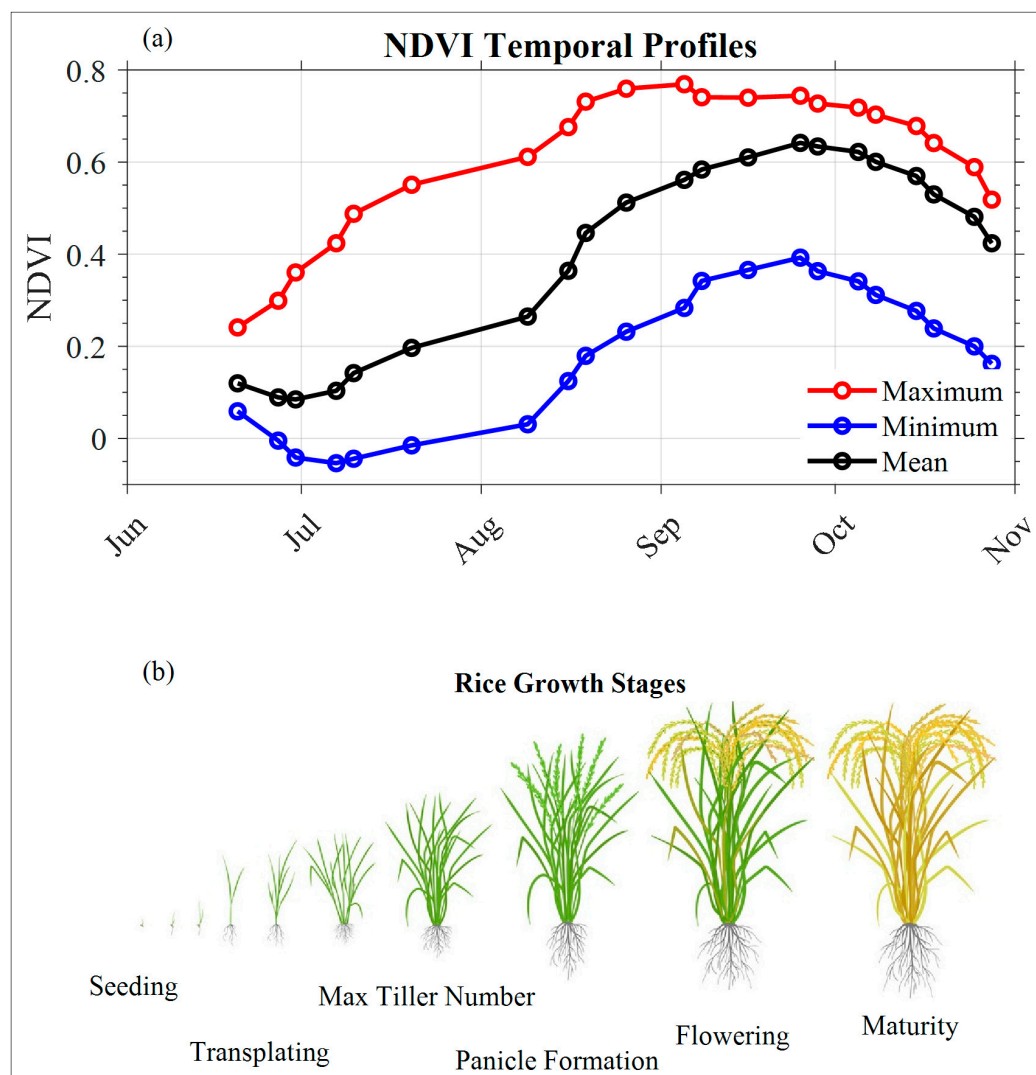

**Figure 5.** (**a**) The temporal profiles of vegetation index (NDVI) and (**b**) different growth stages of rice crop. The points are not equally spaced (subplot **a**) due to the unavailability of cloud free images of the study area (i.e., monsoon season; late July and early August).

### *3.3. Prediction of Rice Yield and the Performance of Vegetation Indices*

The prediction of rice yields using PLSR and time series vegetation indices are summarized Table 3. The integration of PLSR and sequential-time-stamped-vegetation indices were found effective for quantifying rice yields. The time series NDVI yielded the highest $R^2 = 0.83$ (lowest $RMSE_{cv} = 0.12$ ton/ha) followed by EVI ($R^2 = 0.80$, $RMSE_{cv} = 0.14$ ton/ha), SAVI ($R^2 = 0.79$, $RMSE_{cv} = 0.14$ ton/ha), and REP ($R^2 = 0.64$, $RMSE_{cv} = 0.17$ ton/ha). The performance of different indices (NDVI, SAVI, EVI, REP) for rice yield estimation were consistent for both calibration and validation datasets (Table 3, Figure 6c,f,i,l).

**Table 3.** Results of the PLSR applied to time series vegetation indices (NDVI, SAVI, EVI, and REP). Number of latent variables (in PLSR model), calibration $R^2$, validation $R^2$, calibration RMSE ($RMSE_C$), and cross validation RMSE ($RMSE_{CV}$) are summarized.

| Indices | No. of Latent Variables in PLSR Model | Calibration $R^2$ | RMSEC (ton/ha) | Validation $R^2$ | RMSECV (ton/ha) |
|---|---|---|---|---|---|
| NDVI | 6 | 0.87 | 0.11 | 0.83 | 0.12 |
| EVI | 6 | 0.85 | 0.12 | 0.80 | 0.14 |
| SAVI | 6 | 0.84 | 0.12 | 0.79 | 0.14 |
| REP | 5 | 0.70 | 0.16 | 0.62 | 0.17 |

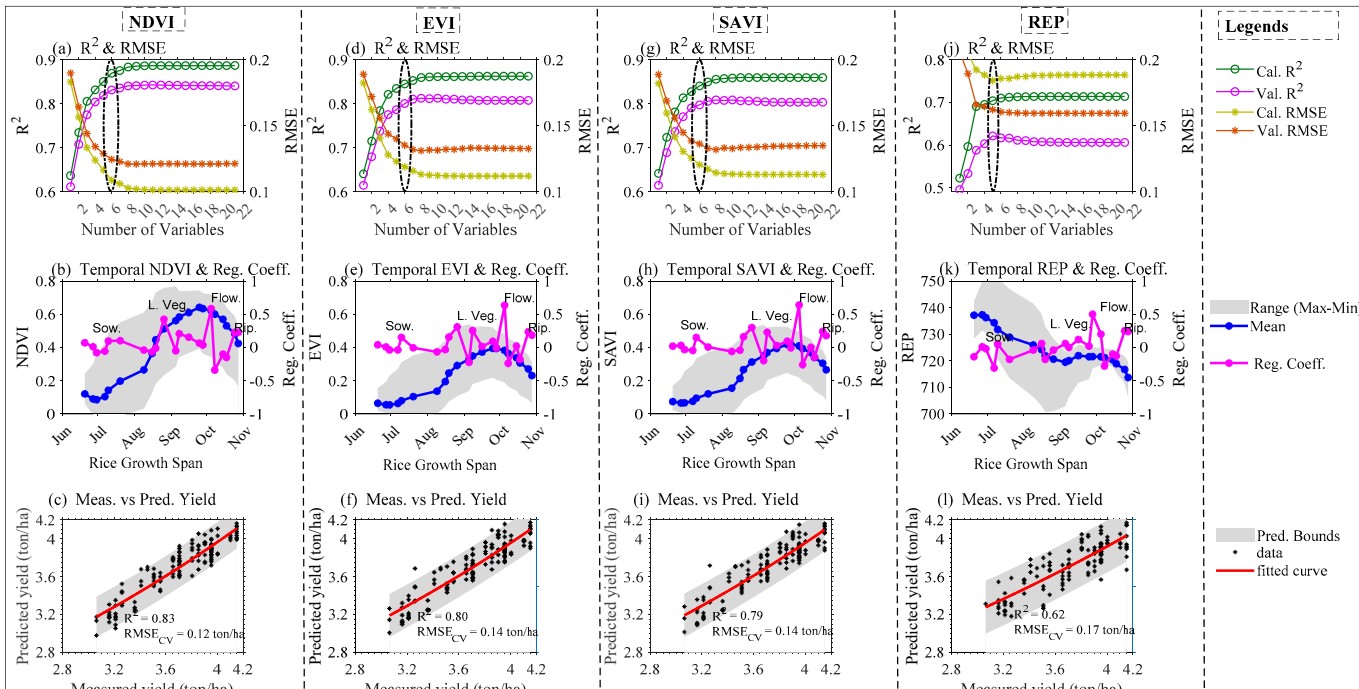

**Figure 6.** The analysis based on temporal NDVI, EVI, SAVI, and REP. The $R^2$ increases and RMSE decreases with augmenting the number of variables unless it reaches saturation (dashed elliptical in Panels **a,d,g,j**). After certain number of latent variables, the decrease in RMSE was negligible and that was taken as the optimum number of variables for PLSR model development. The temporal profiles of vegetation indices are similar in shape except REP (Panels **b,e,h,k**). The regression coefficients lines show that sowing, late vegetative, flowering and ripening are important phases for predicting rice yields using PLSR. The measured vs. predicted (Panels **c,f,i,l**) manifest that yield was best predicted using temporal NDVI data (yielded highest $R^2$ and lowest RMSE).

Using the time series profiles of NDVI, EVI and SAVI, the PLSR models selected six latent variables (Figure 6, Table 3). The selected six latent variables explained most of the variance (e.g., as the case of NDVI where $R^2 = 0.83$) and the addition of further variables hardly improve the model performance (e.g., the total 21 variables yield maximum $R^2$ of 0.84) (Figure 6a,d,g,j). The important latent variable (in this study, time stamped vegetation indices) was located at late vegetative, reproductive (flowering), and ripening phases of rice growth (Figure 6b,e,h,k). Using time series Red Edge Position (REP) data, the number of selected latent variables were five (05) with high regression coefficient (or *B*-coefficient) values at flowering, ripening, and late vegetative phases (Figure 6c).

### 3.4. Spatial Varability in Rice Yield Potential

The distribution map (developed from best predicting PLSR model, time series vegetation indices and map of rice grown area) reflects that rice yield distribution varies in space and ranges between 1.5 to 4.2 ton/ha (Figure 7a). The upper limit of rice yields (i.e., 4.20 ton/ha) in the distribution map were closely matching with the ground measured yield (4.15 ton/ha). The minimum limit was underestimated in the spatial distribution map of rice yield (1.5 ton/ha) compared to the ground measured minimum yield (3.06 ton/ha). The validation of spatial distribution maps against independent ground measured yield data (30% of the total samples) confirms that yield was predicted with high accuracy (see Figure 7b). The high $R^2$ (0.83) and low RMSE (0.14 tons/ha) manifested a close match between measured and predicted rice yields (Figure 7b).

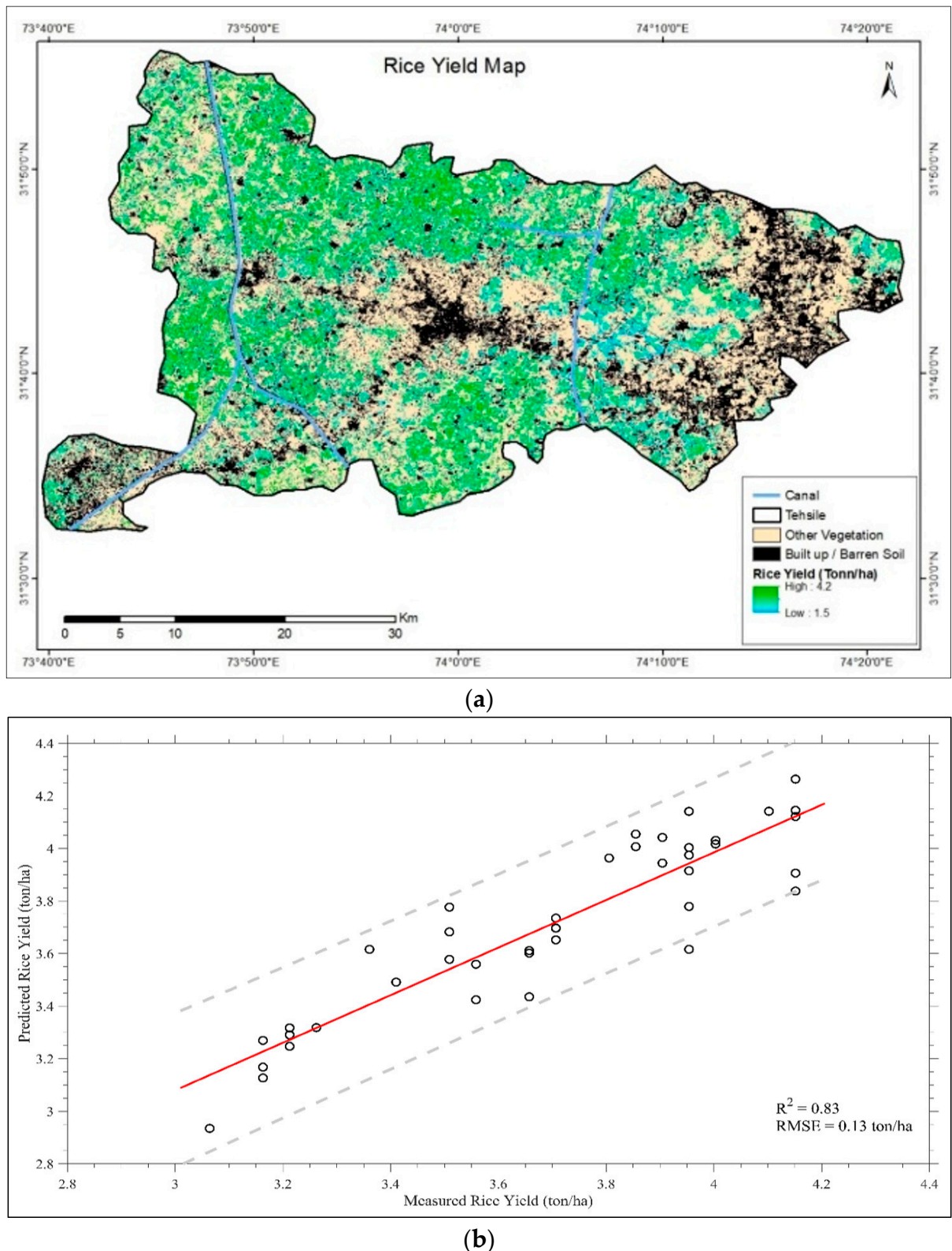

**Figure 7.** The distribution of rice yield varies spatially (ranging from 1.5 to 4.2 ton/ha) in the study area (**a**). The predicted yield (based on distribution map) shows strong relation with measured yield (**b**).

## 4. Discussion

The field data (Table 2) reflect that the measured rice yield production (minimum (3.06 ton/ha), mean (3.7 ton/ha), maximum 4.15 (ton/ha) in the study area is within the limits of rice crop statistics within the country (i.e., ranging from 2.4 ton/ha to 10 ton/ ha). These numbers are far less than the statistics of the neighboring countries (i.e., China, Vietnam, Bangladesh) and could be attributed to the variety of rice [45], uneven water usage, weeds and pest attacks, and post-harvest loses (e.g., shattering and improper drying and storing etc.). The super basmati grown in the study area produces high quality rice and is famous for its aromatic fragrance; however, it is less productive compared to other hybrid varieties.

The time series profiles show that vegetation indices (Figure 8a) are minimum at the transplantation stage and gradually increase in vegetative phase (tillering, panicle, and flowering stages). A decline was observed in the vegetation indices after post flowering phases (e.g., dough, ripening). This initial increase in vegetation index values could be associated with increase in leaf area coverage (LAI, biomass) and the post flowering phase decline could be attributed to the senescing of rice crop. The temporal variation in vegetation indices (in this study) are in line with the findings of previous studies [46,47], where the peak greenness is achieved at flowering/milk phases and decline was observed in dough stages and reaching minimum at ripening phase (Figure 8).

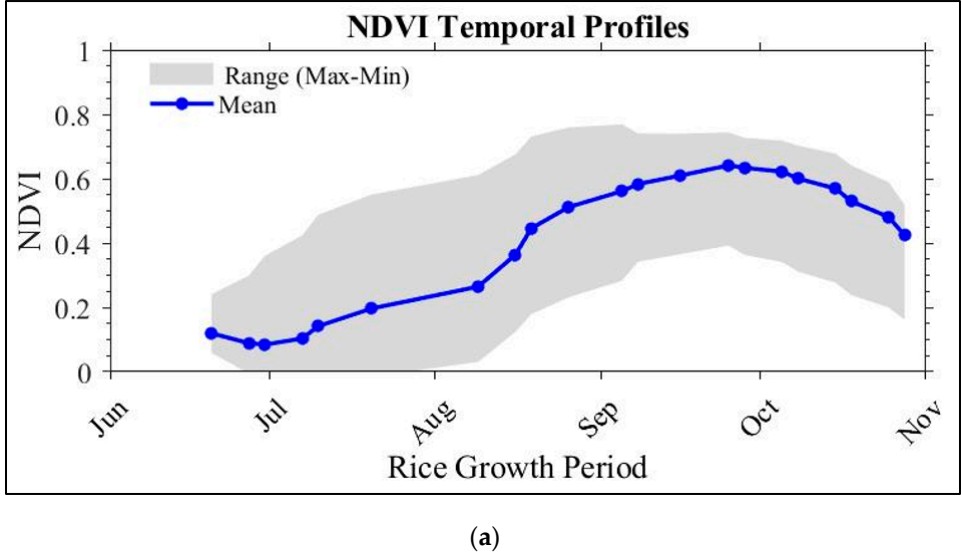

(**a**)

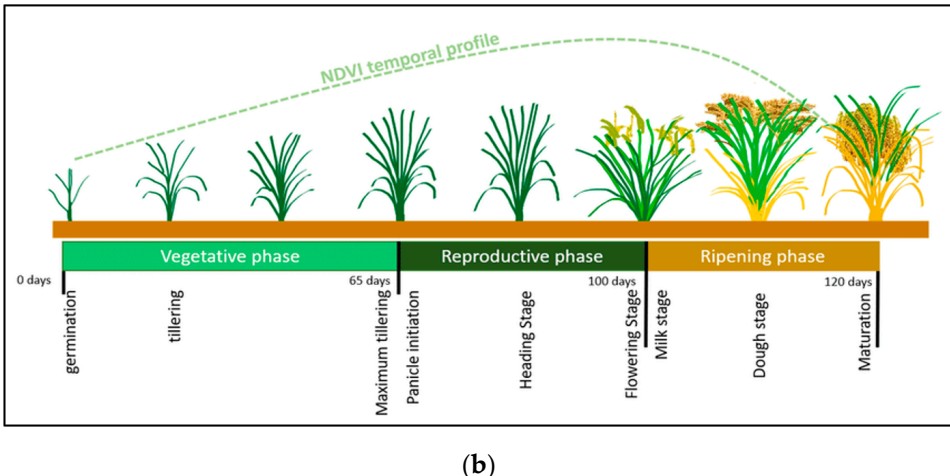

(**b**)

**Figure 8.** (**a**) The range and mean temporal profiles of Normalized Difference Vegetation Index (NDVI) spanning the entire growth period of rice crop. (**b**) The temporal profile of vegetation index (NDVI) is closely in-line with the findings of [47] (shown in plot "**b**").

The integration of PLSR and time series vegetation indices accurately predicted rice yields (the maximum $R^2 = 0.83$ and least RMSE = 0.12 ton/ha (3.12% of the mean yield). The slightly better performance of NDVI (relative to SAVI, EVI, and REP) could be attributed to the canopy characteristics (e.g., structural, vegetation percent cover) of rice crop. The abundance of stem and leaf blades of rice crop obscures the background soil visibility and leads to enhanced vegetation (rice) reflectance signals, thus allowing slope-based-index (such as NDVI) to perform more precise estimates of rice yield [48,49].

Using PLSR regression, the prediction accuracy enhances with augmenting the number of variables (NDVI, EVI, SAVI, and REP) and leads to high $R^2$ and low RMSE until the model stabilized at a certain point. In this study, the PLSR model saturate at the addition of six latent variables (Figure 6a,d,g,j) and captured maximum variance present in dataset. The addition of further variables hardly improves the prediction of rice yield and displays a flat line [50]. The selected latent variables belong to late vegetative, reproductive (panicle, flowering, milky), and ripening phases (Figure 6b,e,h,k) and reflects the critical importance of these growth stages in rice yield production. The picking of late vegetative phase may be associated with an increase in biomass and leaf covered area (leaves and shoots fully develop at this stage and gain maximum crop height). The high correlation at reproductive phases could be associated with the formation panicle, flowering, milk, dough, and maturity of grain, which directly influence the crop yields. The results of this study are also consistent with the findings of existing literature, where booting stage highly influences the rice yield production [51]. The outcomes of this study help in estimating the rice yield and highlight the critical phases in the life cycle (of rice crop) where monitoring and human intervention (such as usage of water and agrochemicals) can enhance the yield production.

## 5. Conclusions

This study aims to accurately quantify rice yield and identify the critical growth stages that influence the rice yield production. The integration of PLSR and time series vegetation indices (i.e., spanning across the entire rice crop growth period) results in accurate predictions of rice yield. Among vegetation indices, NDVI performs the best (yield high $R^2$ and low RMSE) followed by EVI, SAVI, and REP. Using the time stamped vegetation indices, the PLSR coefficients identified the growth stages that influence the rice yield. The growth stages (selected latent variables) belong to late vegetative, reproductive (panicle, flowering, milky), and ripening phases. The selected critical growth stages were common in all four types of vegetation indices (i.e., NDVI, EVI, SAVI, and REP) used. This study concludes that PLSR can effectively be used for rice yield estimation and identifying critical stages of the rice growth cycle. The precise yield estimation (rice in this case) allows decision makers to strategize policy regarding yield import and export. The outcome of this study can also help the farmers to monitor rice at critical time spans and allow them to intervene (e.g., usage of water, fertilizer, pesticides etc.) in a timely manner. The timely interventions thus help in producing more yields which in turn is essential for minimizing hunger (SDG 2), alleviating poverty (SDG 1), ensuring land (SDG 15) and food security.

However, the present study did not compare the accuracy of PLS algorithm with artificial neural networks, support vector machines, other geo-statistics, etc., for yield forecasting. These would be interesting directions for future study.

**Author Contributions:** The study was conceptualization by A.N. and S.U.; methodology, A.A. (Azhar Abbas), M.S.I.; data collection and validation, Z.A.S. and K.H.; formal analysis, M.S. (Muhammad Shakir) and M.S. (Munawar Shah); Planning and Execution, M.U.B.; writing—review and editing, A.N. and A.A. (Asad Ali). All authors have read and agreed to the published version of the manuscript.

**Funding:** We are thankful for the financial support of University of Agriculture Faisalabad (APC funding) and Galaxy Rice Mills Pvt Ltd. Gujranwala for logistics during field data collection.

**Institutional Review Board Statement:** This study is 'Not applicable' for Institutional Review Board and Ethical review as this research work does not involve any humans or animals or biological material.

**Informed Consent Statement:** Not applicable.

**Data Availability Statement:** Data will be made available for research purpose upon request.

**Acknowledgments:** We acknowledge the Copernicus Open Access Hub for providing freely available Sentinel-2 data. We also acknowledge M. Hamid Choudhary, GIS Centre The University of Punjab, Lahore Pakistan for helping in software and Field survey.

**Conflicts of Interest:** The authors declare no conflict of interest.

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
