# Peer review of "Estimation and Forecasting of Rice Yield Using Phenology-Based Algorithm and Linear Regression Model on Sentinel-II Satellite Data"

_agriculture, doi:10.3390/agriculture11101026_

Round 1
Reviewer 1 Report
Timely and accurate rice yield estimates is of great significance for agricultural production management and decision makers. Several issues should be addressed.
- In introduction, the authors just introduced the general methods (field survey, model and remote sensing) for crop yield estimations. More specific information about yield estimation methods should be included. For example, as for remote sensing-based method, what is the widely-used strategy for yield estimation? In addition, More information should be given to explain why did you use the PLSR method.
- In figure 1, how did you get the rice map? What was the classification accuracy of this rice map?
- Photos in figure 2 are not clear enough and are not typical for expressing phenological events.
- Contents in Section 2.2.2 and 2.3 are too redundancy. You can consider to merge them and shorten the contents. In addition, which year did the remote sensing data refer to?
- Flooding has been widely demonstrated as the important characteristic for rice mapping based on remote sensing. Why didn’t you consider the vegetation index that are related to water?
- Section 2.4: Please clarify the specific parameters for the latent variables (LVs)
- Section 2.5: How did crop phonological information matter in generating the rice yield map? Formula R is confusing. How did the “vegetation mask and other land mask” generate? Please clarify them.
- The readability in Materials and Methods section was quite weak. The novelty that using the relationships between rice phenological characteristic and rice yield to estimate yield was not clear in this section.
- In discussion section, the third paragraph mentioned that the three VIs were used to be compared, but the fourth paragraph mentioned that the VIs were integrated for PLSR. But how did the VIs use for PLSR regression that were not described clearly in Method section.
- Some languages mistakes were clear. Languages needs to be further polished.
Reviewer 2 Report
First of all, I want to congratulate the authors for their efforts in this manuscript. They presented an interesting and updated topic, which fits perfectly into the journal scope. The paper is well structured. Nonetheless, some aspects must be improved in order to enhance the quality of the paper. Following, I include a list of those aspects:
- In the introduction, other papers that have used the temporal variability of sentinel images for other purposes should be added to help the readers to see that this methodology is widely used. The authors have added some examples in lines 67-72. Following, I include some references of temporal variability in other issues related to interesting cropping monitoring:
- Basterrechea, D. A., Parra, L., Parra, M., & Lloret, J. (2020, December). A Proposal for Monitoring Grass Coverage in Citrus Crops Applying Time Series Analysis in Sentinel-2 Bands. In International Conference on Industrial IoT Technologies and Applications (pp. 193-206). Springer, Cham.
- At the end of the introduction, add a paragraph in which the aim or objectives of the paper are detailed. If possible, in this paragraph, describe the main novelty and highlight the relevance of the results.
- Subsection 2.2 detail the included bands in the study. Moreover, add the spatial resolution of selected bands.
- In Table 2, I suggest including the kurtosis values and asymmetry of the distribution of both subsamples.
- Figure 6 should be reshaped. It is not possible to read it. I suggest, if possible, use a horizontal page to put this Figure. The Figure is fascinating, and readers would like to analyze it in detail.
- At the end of the conclusions, authors must include the future work linked to their results and discussions.
Minor issues:
The caption of figures should be placed below the Figure. Check and correct it in Figure 1 and check other cases.
Check the format of eq 2
Round 2
Reviewer 1 Report
The revised paper has improved a lot. I recommonded accept in present form.